# SRA: A Novel Method to Improve Feature Embedding in Self-supervised Learning for Histopathological Images

**Hamid Manoochehri**[*1,2] (iD)                    HAMID.MANOOCHEHRI@UTAH.EDU
[1] *Scientific Computing and Imaging Institute, University of Utah, Salt Lake City, UT, USA*
[2] *Dept. of Electrical and Computing Engineering, University of Utah, Salt Lake City, UT, USA*

**Bodong Zhang**[*†1,2] (iD)                    BODONG.ZHANG@UTAH.EDU
**Beatrice S. Knudsen**[3] (iD)          BEATRICE.KNUDSEN@PATHOLOGY.UTAH.EDU
[3] *Department of Pathology, University of Utah, Salt Lake City, UT, USA*

**Tolga Tasdizen**[1,2] (iD)                    TOLGA.TASDIZEN@UTAH.EDU

**Editors:** Accepted for publication at MIDL 2025

## Abstract

Self-supervised learning has become a cornerstone in various areas, particularly histopathological image analysis. Image augmentation plays a crucial role in self-supervised learning, as it generates variations in image samples. However, traditional image augmentation techniques often overlook the unique characteristics of histopathological images. In this paper, we propose a new histopathology-specific image augmentation method called stain reconstruction augmentation (SRA). We integrate our SRA into various self-supervised learning models. We demonstrate that our SRA always improves the standard models across various downstream tasks and achieves superior performance to a state-of-the-art foundation model pre-trained on significantly larger histopathology datasets.

**Keywords:** image augmentation, histopathological image, self-supervised learning, contrastive learning

## 1. Introduction

In this paper, we propose a novel image augmentation method on H&E stained histopathological images called stain reconstruction augmentation (SRA). Through experiments, we demonstrate that our SRA further improves the original self-supervised learning (SSL)(Chen et al., 2021; Caron et al., 2021) models based on performances in various downstream tasks.

## 2. Stain Reconstruction Augmentation (SRA)

As shown in Figure 1, for an H&E image, we first perform stain separation using the algorithm from (Macenko et al., 2009) to obtain single-stain images. For each whole slide image (WSI), the RGB pixel values are mapped into Optical Density (OD) space $(OD_R, OD_G, OD_B)$ according to the Beer-Lambert law (Beer, 1852; Lambert, 1760), where higher OD values indicate stronger stains:

$$OD_C = \log_{10}(I_{0,C}/I_C) \tag{1}$$

---

[*] Contributed equally

[†] Corresponding author

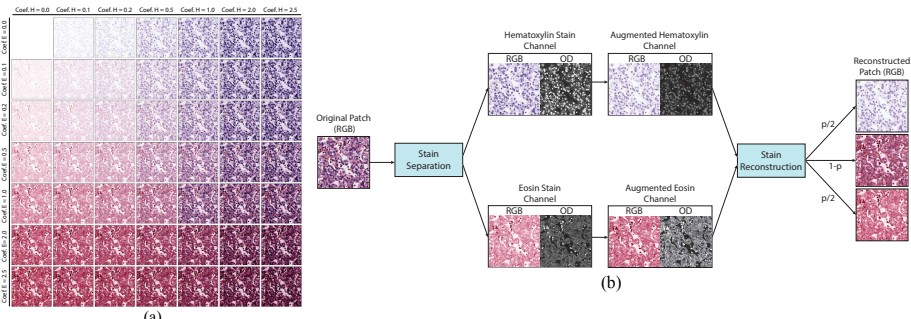

Figure 1: (a) Examples of augmentations by SRA with different target strengths of H channel and E channel (b) Pipeline of stain reconstruction augmentation (SRA).

The channel C is red, green, or blue channel. The $I_{0,C}$ denotes background intensity, which is usually 255. $I_C$ is the intensity in channel C in current pixel. All pixels are mapped to the same OD space. Based on the distribution of these pixels in OD space, three unit vectors, $V_H$, $V_E$, and $V_{Residual}$, are derived, which allow for the decomposition of OD values:

$$(OD_R, OD_G, OD_B) = \alpha V_H + \beta V_E + \gamma V_{Residual} \qquad (2)$$

In each slide, we calculate the values of $\alpha$ and $\beta$ for each tissue pixel, where $\alpha$ represents the proportion of Hematoxylin stain and $\beta$ represents the proportion of Eosin stain for each pixel. For each WSI, we define $H_{max}$ as the maximum intensity of all $\alpha$ values in the tissue pixels of the slide and similarly define $E_{max}$ for the maximum intensity for $\beta$.

In the first step of implementing our SRA, we predefine global target ranges for each stain after augmentation. For each training image, following the stain separation process, we independently and randomly select coefficients $coef_H$ and $coef_E$ from within the target ranges. We then multiply $\alpha$ by $coef_H/H_{max}$ and $\beta$ by $coef_E/E_{max}$ to randomly adjust the stain strength. Furthermore, inspired by the previous work of contrastive learning between pure Hematoxylin images and pure Eosin images, (Zhang et al., 2022a, 2023) we introduce a hyperparameter $p$, which defines the probability of randomly setting either $coef_H$ or $coef_E$ to zero during stain reconstruction augmentation, thereby creating additional variations. Finally, after all processes are complete, we reconstruct the images back into RGB space from OD space, based on the new proportions of Hematoxylin and Eosin stains. Our SRA is an unsupervised method that does not use any labels to apply the algorithm.

## 3. Experiments

We conducted our experiments on three datasets: The Cancer Genome Atlas Kidney Renal Clear Cell Carcinoma (TCGA KIRC) dataset (National Cancer Institute, 2023), the Utah KIRC dataset (Zhang et al., 2023), and the Utah Renal Vein Thrombus (RVT) dataset (Zhang et al., 2025). The TCGA KIRC dataset and Utah KIRC dataset provide 300 slides (cropped to 1,646,665 patches) and 32 slides (cropped to 208,291 patches), separately, for self-supervised learning, as well as additional patches with patch-level labels for downstream patch classification tasks. The Utah RVT dataset is only used for multiple instance learning

in downstream tasks. We ran each setting 3 times to calculate the mean and standard deviation. (More details about datasets can be found in Appendix C.1)

We first evaluated SRA by performing self-supervised pre-training and downstream classification on the same datasets (TCGA KIRC or Utah KIRC). Results in Table 1 show that our SRA greatly enhances original MoCo v3 (Chen et al., 2021) during pre-training. We call the new model SRA-MoCo v3. For reference, we also compared the classification results with a state-of-the-art foundation encoder model - Prov-GigaPath (Xu et al., 2024).

| Model | TCGA KIRC (20X) | | Utah KIRC (10X) | |
|---|---|---|---|---|
| | Pre-trained Dataset | Balanced Accuracy (%) | Pre-trained Dataset | Balanced Accuracy (%) |
| ResNet50 | ImageNet | 69.97 ± 5.59 | ImageNet | 87.76 ± 0.10 |
| Prov-GigaPath | >170,000 Slides >1,000,000,000 patches | 77.88 ± 1.22 | >170,000 Slides >1,000,000,000 patches | 95.28 ± 1.02 |
| MoCo v3 | TCGA KIRC | 79.37 ± 1.18 | Utah KIRC | 93.77 ± 0.86 |
| MoCo v3 + TSA | 300 Slides | 81.50 ± 0.23 | 49 Slides | 94.00 ± 0.26 |
| SRA-MoCo v3 | 1,646,665 patches | **83.62** ± 0.28 | 208,291 patches | **95.85** ± 0.34 |

Table 1: Comparison of pre-trained models' performance on TCGA KIRC (20X magnification) and Utah KIRC (10X magnification). TSA means traditional stain augmentation methods (Tellez et al., 2018a,b). (Differences are shown in Appendix B.)

To evaluate in a transfer learning setting, we pre-trained encoders on the TCGA KIRC dataset and subsequently evaluated the encoders on the Utah KIRC and the Utah RVT. Based on Table 2, we observed 2.8% improvement on Utah KIRC and 3.7% improvement on Utah RVT with SRA-MoCo v3 compared to MoCo v3 on balanced accuracy.

| Model | Utah KIRC Balanced Acc. | Utah RVT Balanced Acc. | Utah RVT F1-score | Utah RVT AUC |
|---|---|---|---|---|
| Prov-GigaPath | 95.28 ± 1.02 | 70.96 ± 1.40 | 64.77 ± 1.66 | 0.7576 ± 0.0101 |
| DINO (TCGA KIRC) | 95.49 ± 0.24 | 67.51 ± 1.96 | 62.54 ± 1.29 | 0.7206 ± 0.0154 |
| SRA-DINO (TCGA KIRC) | 96.37 ± 0.21 | 69.70 ± 1.01 | 63.98 ± 0.33 | 0.7323 ± 0.0152 |
| MoCo v3 (TCGA KIRC) | 95.32 ± 0.30 | 71.80 ± 2.12 | 66.51 ± 2.80 | 0.7677 ± 0.0307 |
| MoCo v3 + TSA (TCGA KIRC) | 94.17 ± 0.82 | 73.40 ± 3.36 | 67.74 ± 3.26 | 0.7609 ± 0.0776 |
| SRA-MoCo v3 (TCGA KIRC) | **98.12** ±0.15 | **75.50** ±7.08 | **71.11** ± 7.70 | **0.8013** ± 0.0337 |

Table 2: Performance of pre-trained models on Utah KIRC and Utah RVT dataset.

Lastly, we conducted ablation studies to analyze the impact of hyperparameter settings in SRA. Based on the experimental results (in Appendix C.3), the best performance was achieved with target ranges of [0.1, 2.5]. Both narrower and wider ranges negatively affected the performance of the SRA model. Even though effective, if the possibility p that only a single channel is adopted is too large, it produces a harmful effect on the training. In experiments, we abandoned the residual part during stain reconstruction as keeping residual part does not provide benefits. We also found that both SRA and standard color augmentations are beneficial, so we kept both of them in SRA-MoCo v3. We made our SRA-MoCo v3 code publicly available at github.com/hamidmanoochehri/Paper_SRA

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

## Appendix A. Introduction of related models

### A.1. Self-supervised learning and contrastive loss

Recent advancements in self-supervised learning (Zbontar et al., 2021; Grill et al., 2020) have introduced novel frameworks for learning robust and accurate features across various datasets. Barlow Twins (Zbontar et al., 2021) encourages two augmented views of the same input to produce similar but decorrelated representations by minimizing the cross-correlation between them. DINO (Caron et al., 2021; Oquab et al., 2024; Darcet et al., 2024) employs a student-teacher framework within a vision transformer architecture, without requiring labeled data. In DINO, the student encoder attempts to mimic the teacher encoder, which is updated based on an exponential moving average (EMA). Unlike the

student encoder, which processes both global and local views of the images, the teacher encoder only receives global views. Building on the DINO framework, PathDino (Alfasly et al., 2024) combines lightweight transformers with a novel 360° rotation augmentation (HistoRotate), achieving robust performance across 12 diverse pathology datasets.

Contrastive learning is one of the most widely used and fundamental approaches in self-supervised learning pipelines. For instance, the SimCLR framework (Chen et al., 2020b,a) utilizes NT-Xent loss on strongly augmented views of images, aiming to minimize the distance between different views of the same image while maximizing the distance between views of different images. In contrast, SwAV (Caron et al., 2020) employs a cluster-based contrastive learning approach rather than a pairwise one, using a swapped prediction mechanism to encourage the features of the same cluster to be as invariant as possible. Other popular contrastive learning methods include iBOT (Zhou et al., 2022), RePre (Wang et al., 2022) and RECON (Qi et al., 2023).

### A.2. Pathology-specific augmentation

In the context of pathology-specific augmentations, various methods have been proposed to address domain-specific challenges and variations in histopathological images. Shen et al. (Shen et al., 2022) introduced RandStainNA, which generates random template slides for color normalization and augmentation in HSV, LAB, and HED color spaces to tackle variations in staining and colors across different slides and datasets. Additionally, Gullapally et al. (Gullapally et al., 2023) addressed inter-laboratory and scanner variability through Scanner Transform (ST) and Stain Vector Augmentation (SVA), enhancing out-of-distribution performance on tasks such as tissue segmentation.

A fundamental operation for many pathology-specific augmentations is stain separation, which isolates single-channel images in Optical Density (OD) space based on the Beer-Lambert law (Lambert, 1760; Beer, 1852). In (Yang et al., 2022), perturbations are applied to the stain separation matrix to deal with the errors in separation matrix calculation. In (Tellez et al., 2018a,b), each channel is randomly scaled and biased within a narrow range before converting back to RGB space. However, the maximum possible intensity after augmentation is still influenced by the original image's maximum intensity. (Chang et al., 2021) also utilizes this method, along with random stain matrix interpolation, to handle domain variations across datasets by incorporating information from both source and target data.

## Appendix B. Integration of SRA into MoCo v3 (An overview of SRA-MoCo v3)

Figure 2 shows the overall workflow of SRA-MoCo v3. First, all pixels within the tissue regions of an H&E whole slide image (WSI) are collected to analyze the max intensity (strength) of each stain, where the intensity is measured on Optical Density (OD) space after stain separation (Ruifrok and Johnston, 2001) process. To perform stain reconstruction augmentation, we predefine an absolute range for the target strength of each stain and map the real strength of each stain to a random value in this target range. Unlike the approach in (Tellez et al., 2018a,b; Chang et al., 2021), which only slightly adjusts the strength of each stain within a relative range by multiplying a random factor between 95% and 105%,

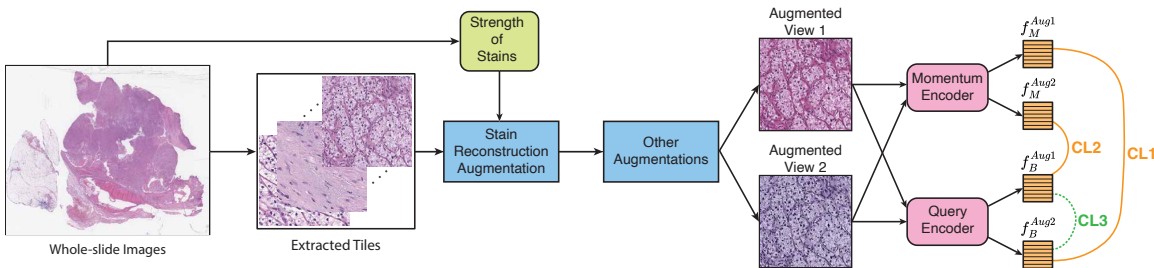

Figure 2: The pipeline of our SRA-MoCo v3. We integrate our Stain Reconstruction Augmentation (SRA) as well as additional contrastive loss term (CL3) into MoCo v3. The f in the figure shows the output features from the encoders.

where the augmented images are statistically affected by the intensities of original stain channel images, our SRA directly defines a much broader absolute range for target strength of each stain channel. For instance, if the target range is set to [0.1, 2.5], and the original strength of a particular stain in a WSI is 2.5, which indicates a deeply stained image. Augmenting this stain channel only makes the new maximum intensity fall inside [0.1, 2.5], without surpassing the original strength since original image is already a deepest stained image in our defined range. While the traditional augmentation makes the new maximum intensity fall between 2.375 and 2.625. Our method allows for more extensive and stronger augmentations while ensuring the strength remains within an appropriate range. Moreover, inspired by multi-modal contrastive learning (Chai and Wang, 2022; Zhang et al., 2022a), we create probability p for excluding one stain channel, allowing only the other stain channel to remain after augmentation. Figure 3 shows the differences between traditional stain augmentation and our SRA.

After stain reconstruction augmentation, additional general image augmentation methods are applied to introduce further variations. MoCo v3 is used as the backbone for histopathology image representation learning. In MoCo v3, the contrastive loss is computed between queries from the query encoder and keys from the momentum encoder using different augmentations. However, there are no loss term that specifically focus on contrastive learning between different augmentations from the same query encoder. Given the substantial variations introduced by stain reconstruction augmentation, we further explored the addition of this contrastive loss term that focuses solely on augmentations. This additional loss was only designed for query encoder on MoCo framework, as there is no gradient's backpropagation on momentum encoder.

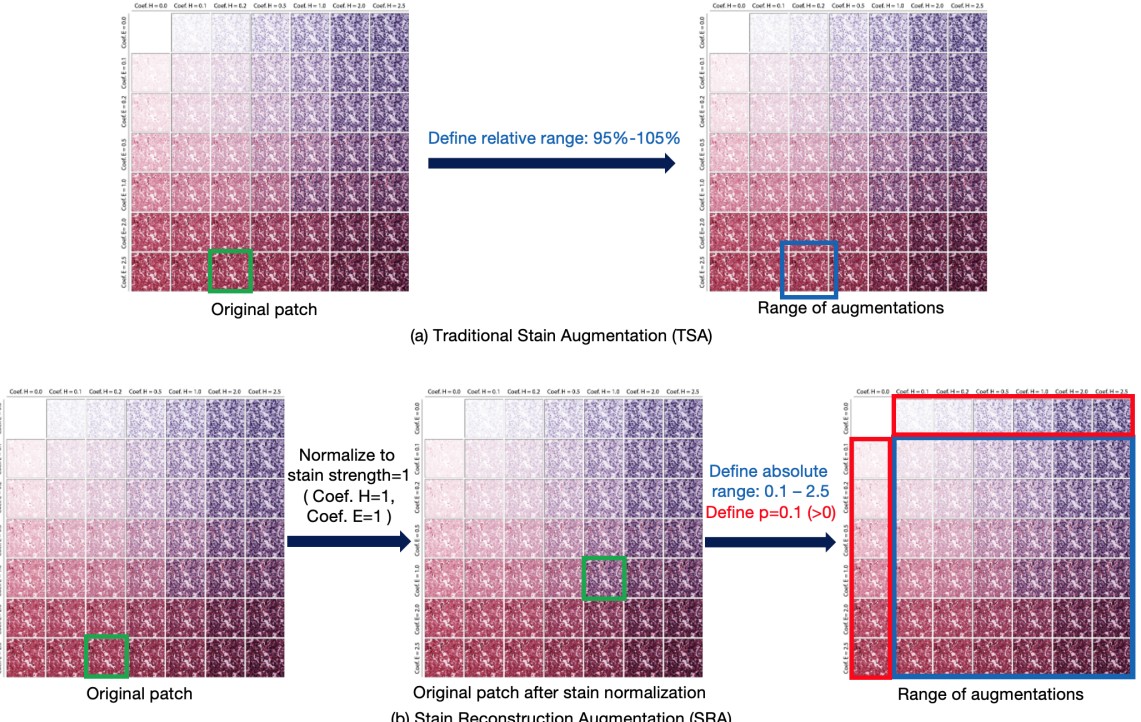

Figure 3: Comparisons between traditional stain augmentation (TSA) and our stain reconstruction augmentation (SRA). The blue rectangles show the range of augmentations. If the probability p for excluding one stain channel is not zero, then SRA further introduces new augmentation results shown in red rectangles.

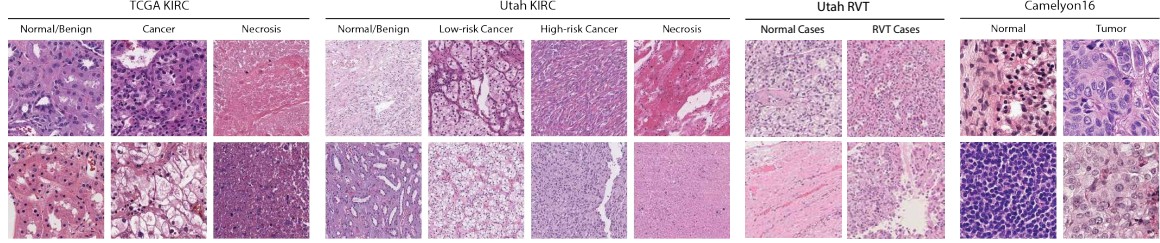

Figure 4: Patch examples from different classes and different datasets.

## Appendix C. Additional information in experiments

### C.1. Datasets details

We conducted all main experiments [1] on three datasets: The Cancer Genome Atlas Kidney Renal Clear Cell Carcinoma (TCGA KIRC) dataset (National Cancer Institute, 2023),

---

1. The experiments were funded by NIH/NCI R21CA277381, DoD HT94252410186, and Department of Veterans Affairs I01CS002622. Part of the results presented in this paper are based upon data generated by the TCGA Research Network: https://www.cancer.gov/tcga

the Utah KIRC dataset (Zhang et al., 2023), the Utah Renal Vein Thrombus (RVT) dataset(Zhang et al., 2025), all of which consist of H&E stained kidney WSIs. In ablation studies, we also include the Camelyon 16 breast cancer dataset (Ehteshami Bejnordi et al., 2017, 2023). Patch examples are shown in Figure 4.

The TCGA KIRC dataset includes 420 WSIs in total, with 300 slides used for self-supervised pre-training. The 300 slides provide 1,646,665 tissue patches of size 400x400 if cropped at 20X resolution. Patch-level labels are also provided in the remaining slides for downstream patch-level 3-class classification tasks. More details are provided at Table 3.

In the Utah KIRC dataset, there are 49 slides from different patients, with 32 slides for self-supervised pre-training. In total, the 32 slides provide 208,291 tissue patches of size 400x400 at 10X resolution. During downstream 4-class patch classification tasks, the 32 slides also provide a small portion of patches that have patch-level labels. The remaining 17 slides provide patches with labels for validation and test. More details can be found at Table 3.

| Task | | TCGA KIRC | | | | Utah KIRC | | | | |
|---|---|---|---|---|---|---|---|---|---|---|
| **SSL** | | Total Unlabeled Patches 1,646,665 | | | | Total Unlabeled Patches 208,291 | | | | |
| **Downstream Classification (Evaluation)** | Set | Normal/ Benign | Cancer | Necrosis | Total | Normal/ Benign | Low-Risk Cancer | High-Risk Cancer | Necrosis | Total |
| | Train | 84,578 | 180,471 | 7,932 | 272,981 | 28,497 | 2,044 | 2,522 | 4,115 | 37,178 |
| | Val. | 19,638 | 79,382 | 1,301 | 100,321 | 5,472 | 416 | 334 | 2,495 | 8,117 |
| | Test | 15,323 | 62,565 | 6,168 | 84,056 | 7,263 | 598 | 389 | 924 | 9,174 |

Table 3: Summary of the number of patches for each category in each set on the TCGA and Utah KIRC datasets. The training sets include labeled and unlabeled patches.

The Utah RVT dataset is for multiple instance learning (MIL) tasks. Instead of slide-level labels in most of MIL tasks, only case-level (patient-level) labels are provided, while a case could contain multiple slides. In summary, there are 74 negative cases and 31 positive cases of Renal Vein Thrombus in the training set, with 12 negative cases and 8 positive cases in the validation set, as well as 18 negative cases and 11 positive cases in the test set. The total number of slides in training set is 862. The total number of patches in the training set is 2,214,311. In experiments, the patch features are generated by encoders pre-trained on TCGA KIRC dataset to evaluate different self-supervised learning methods in a transfer learning setting.

Camelyon 16 is another public dataset consisting of breast cancer slides for multi-instance learning. It provides only slide-level labels for binary classification between Normal and Tumor. The training set contains 159 Normal slides and 111 Tumor slides, while the test set includes 80 Normal slides and 49 Tumor slides. We randomly selected 10% of the training slides to create a validation set. We followed the standard procedure as other papers (Zhang et al., 2022b) to crop patches in the 20X resolution. Similar to Utah RVT dataset, the patch features are also generated by encoders pre-trained on TCGA KIRC dataset in experiments.

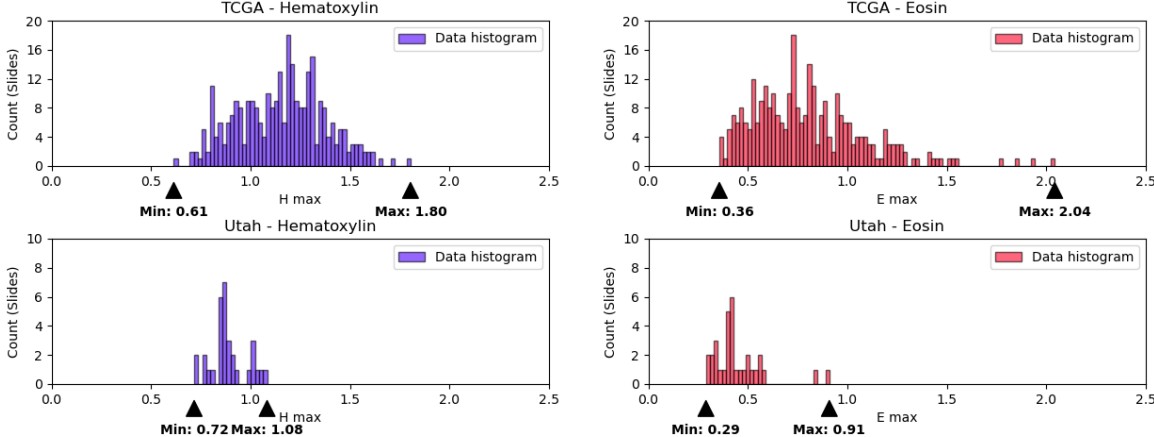

Figure 5: Distributions of strengths of Hematoxylin stain and Eosin stain in Optical Density (OD) space on TCGA training set and Utah training set.

## C.2. Experiment settings

In experiments, we began by pre-training encoders using self-supervised feature representation learning, followed by testing the pre-trained encoders on downstream tasks. Before pre-training, we performed a one-time operation to calculate stain separation matrix for each slide following the code from `https://github.com/BzhangURU/Paper_CLASS-M/tree/main/Section2_get_stain_separation_matrices`. The computational overhead of SRA during pre-training is negligible.

For stain reconstruction augmentation, we first predefined a range for the target strengths of the Hematoxylin and Eosin stains. We calculated the distributions of $H_{max}$ and $E_{max}$ across all training slides in both TCGA KIRC and Utah KIRC datasets. As shown in Figure 5, the stain strengths $H_{max}$ and $E_{max}$ varies across different slides in both datasets. In our experiments, we mainly tested two sets of target ranges. The first set marginally covers the distribution of stain intensities in TCGA KIRC, with a target range of $[0.5, 2.0]$ for new $H_{max}$ and $[0.2, 2.0]$ for new $E_{max}$. The second set has wider target ranges to introduce stronger augmentations, with both new $H_{max}$ and new $E_{max}$ set to $[0.1, 2.5]$ after discussion with pathologists. We selected this range because further widening would cause saturation when reconstructing images from Optical Density (OD) space back to RGB space.

After applying stain reconstruction augmentation, we followed the standard MoCo v3 training procedure with batch size as 512, including additional general augmentations, with ResNet50 (He et al., 2015) as backbone. We also included an option in our code to add the augmentation contrastive loss, $CL_{aug}$, as mentioned before. For the downstream tasks on the Utah RVT dataset and Camelyon 16 dataset, we employed DTFD-MIL (Zhang et al., 2022b). Stain reconstruction augmentation was only applied during pre-training, not in any downstream tasks. All experiments were conducted using Python 3.11.4, PyTorch 2.0.1, torchvision 0.15.2, and CUDA 11.8 on NVIDIA RTX A6000 GPUs.

| Range $coef_H$ | Range $coef_E$ | $p$(only H or E) | Extra Loss | Balanced Acc. Utah KIRC | F1-score Camelyon16 | Accuracy Camelyon16 | Balanced Acc. Camelyon16 |
|---|---|---|---|---|---|---|---|
| N/A | N/A | 0 | — | $95.32 \pm 0.30$ | $80.75 \pm 1.35$ | $85.79 \pm 1.79$ | $84.33 \pm 0.89$ |
| N/A | N/A | 0 | $CL_{aug}$ | $95.34 \pm 0.41$ | $84.57 \pm 2.53$ | $88.11 \pm 1.19$ | $87.78 \pm 2.44$ |
| [0.5, 2.0] | [0.2, 2.0] | 0 | — | $96.51 \pm 0.37$ | $82.44 \pm 0.47$ | $87.34 \pm 0.45$ | $85.58 \pm 0.42$ |
| [0.1, 2.5] | [0.1, 2.5] | 0 | — | $96.95 \pm 0.76$ | $84.94 \pm 1.54$ | $89.67 \pm 0.89$ | $87.19 \pm 1.25$ |
| [0.5, 2.0] | [0.2, 2.0] | 0 | $CL_{aug}$ | $96.86 \pm 0.17$ | $83.41 \pm 0.60$ | $88.37 \pm 0.78$ | $86.15 \pm 0.33$ |
| [0.1, 2.5] | [0.1, 2.5] | 0 | $CL_{aug}$ | $98.09 \pm 0.12$ | $85.96 \pm 2.13$ | $89.15 \pm 2.05$ | $88.75 \pm 1.40$ |
| [0.5, 2.0] | [0.2, 2.0] | 10% | $CL_{aug}$ | $97.41 \pm 0.08$ | $90.79 \pm 1.50$ | $93.28 \pm 1.18$ | $92.08 \pm 1.02$ |
| [0.1, 2.5] | [0.1, 2.5] | 10% | $CL_{aug}$ | $\mathbf{98.12} \pm 0.15$ | $\mathbf{92.07} \pm 0.84$ | $\mathbf{94.31} \pm 0.44$ | $\mathbf{92.91} \pm 0.95$ |

Table 4: Ablation study results showing the impact of each component of SRA-MoCo v3 on Utah KIRC dataset and Camelyon 16 dataset.

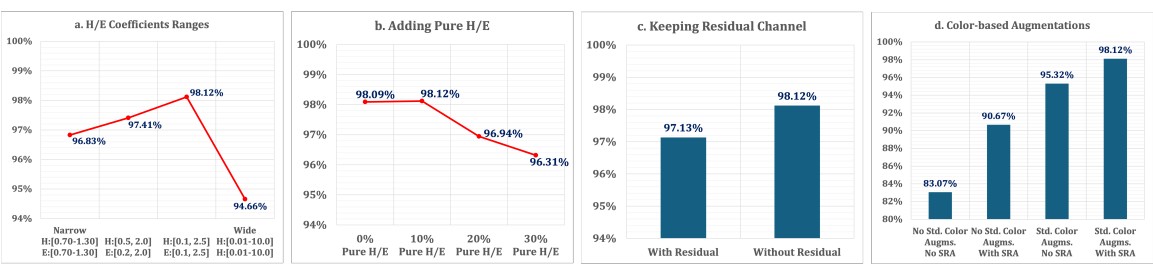

Figure 6: Systematic analysis of hyperparameters of SRA on Utah KIRC dataset.

## C.3. Ablation studies

We also conducted ablation studies to analyze the impact of components in SRA-MoCo v3.

The ablation studies for transfer learning from the TCGA KIRC dataset to the Utah KIRC dataset are presented in Table 4. From the results, we found that simply adding the extra augmentation contrastive loss to MoCo v3 does not yield any improvement. However, this loss becomes effective when combined with stain reconstruction augmentation. Using a wider target range for stain strength and incorporating the possibility of generating pure Hematoxylin or pure Eosin images in stain reconstruction augmentation also proved beneficial. In summary, SRA-MoCo v3 addresses domain shift by generating highly augmented images that are also clinically meaningful. The Figure 6 more systematically analyzed the impact of hyperparameters' selections to the performance. Figure 6(a) shows that target ranges that are too narrow or too wide both have negative impacts to the SRA model. According to Figure 6(b), if the possibility p that only a single channel is adopted is too large, it produces a harmful effect on the training. In experiments, we abandoned the residual part during stain reconstruction. As shown in Figure 6(c), keeping residual part does not provide benefits. In Figure 6(d), we tried to remove standard color-related augmentations like color jitterring. We found that both SRA and standard color augmentations are beneficial, so we kept both of them in SRA-MoCo v3.

Lastly, we evaluated the contribution of each component in SRA-MoCo v3 through downstream tasks on the Camelyon 16 dataset. We observed that both the augmentation

contrastive loss and stain reconstruction augmentation independently improve performance. When combining the augmentation contrastive loss with stain reconstruction augmentation, we achieved a 4.42% improvement in balanced accuracy. Also, we observed a 4.16% increase in balanced accuracy on Camelyon 16 by simply adjusting p from 0 to 0.1, which implies that stronger augmentations are more beneficial on cases with stronger domain shift.

