# OpenReview forum: "SRA: A Novel Method to Improve Feature Embedding in Self-supervised Learning for Histopathological Images"
_MIDL.io/2025/Short_Papers — MIDL 2025 - Short Papers_

### Official Review · Reviewer_dNxt · 2025-04-25

**Rating:** 5
**Confidence:** 5

**Summary:**

This paper introduces Stain Reconstruction Augmentation (SRA), a novel approach for histopathological image augmentation that operates through H-channel and E-channel data enhancement. The authors employ these augmented sample pairs to conduct contrastive learning. Their approach demonstrates superior performance compared to other state-of-the-art methods in downstream fine-tuning and transfer learning tasks. By leveraging the unique characteristics of histopathological images, SRA enables more effective self-supervised learning in medical image analysis.

**Strengths:**

1. Well-structured presentation: The manuscript exhibits excellent organization with comprehensive background introduction, detailed methodological explanations, and clearly articulated contributions, facilitating reader comprehension of the proposed approach.

2. Technical innovation in domain-specific augmentation: The proposed SRA technique leverages the intrinsic correlations between H and E channels in histopathological slides, implementing a random channel erasure strategy that extends existing stain invariance approaches.

3. Comprehensive experimental validation: The evaluation is thorough and convincing, encompassing both fine-tuning and transfer learning configurations. The authors compare their method against multiple pre-trained models with various architectures, providing a comprehensive assessment of SRA's capabilities in self-supervised learning for histopathological images. The extensive ablation studies effectively demonstrate the efficacy of both SRA and its enhanced contrastive loss.

**Weaknesses:**

1. Inadequate comparative analysis with TSA: While the manuscript presents a novel approach, it lacks sufficient discussion comparing SRA with traditional stain augmentation (TSA) techniques. A more detailed analysis of their technical distinctions would strengthen the paper's positioning within the existing literature and better highlight SRA's specific advantages over conventional methods.

---

### Decision · Program_Chairs · 2025-05-01

Accept